# Impact of COVID-19 on health-related quality of life in the general population: A systematic review and meta-analysis

**Desire Aime Nshimirimana**[1,2,3]*, **Donald Kokonya**[4], **Jesse Gitaka**[1], **Bernard Wesonga**[4], **Japheth Nzioki Mativo**[5], **Jean Marie Vianey Rukanikigitero**[6]

1 Departement of Research & Innovation, Mont Kenya University, Thika, Kenya, 2 Department of Health Systems Management, School of Health Sciences, Nairobi Campus, Kenya Methodist University (KeMU), Meru, Kenya, 3 College of Doctoral Studies, Grand Canyon University, Phoenix, Arizona, United States of America, 4 Department of Community Health & Behavioral Sciences, School of Medicine, Masinde Muliro University of Science & Technology, Kakamega, Kenya, 5 Department of Environmental Health, Colleges of Health Sciences, Jumeira University, Dubai, The United Arab Emirates, 6 Department of Dermatology, International Hospital Kampala, Kampala, Uganda

* desireaime.da@gmail.com

**Data Availability Statement:** A protocol was registered in the international Prospective Register

## Abstract

The World Health Organization declared coronavirus disease of 2019 as an epidemic and public health emergency of international concern on January 30th, 2020. Different factors during a pandemic can contribute to low quality of life in the general population. Quality of life is considered multidimensional and subjective and is assessed by using patient reported outcome measures. The aim and objective of this review is to assess the impact of coronavirus disease of 2019 and associated factors on the Quality of Life in the general population. This review was conducted and reported according to the Preferred Reporting Items for Systematic Reviews and Meta-Analyses. A protocol was registered in the international Prospective Register of Systematic Reviews database(CRD42021269897). A comprehensive electronic search in PubMed, EBSCO Host Research Databases, MEDLINE and Google scholar search engine was conducted. A total number of 1,7000,074 articles were identified from electronic search. 25 full text articles were retained for qualitative synthesis and seventeen articles for quantitative analysis. Seven main quality of life scales were used to assess the quality of life of the general population; World Health Organization Quality of Life-bref, EuroQuality of Life-Five dimensions, Short Form, European Quality of Life Survey, coronavirus disease of 2019 Quality of Life, General Health Questionnaire12 and My Life Today Questionnaire. The mean World Health Organization Quality of Life-brief was found to be 53.38% 95% confidence interval [38.50–68.27] and EuroQuality of Life-Five dimensions was 0.89 95% confidence interval [0.69–1.07]. Several factors have been linked to the Coronavirus disease of 2019 such as sociodemographic factors, peoples living with chronic diseases, confinement and financial constraints. This review confirms that the Coronavirus disease of 2019 pandemic affected the quality of life of the general population worldwide. Several factors such as sociodemographic, peoples living with chronic diseases, confinement and financial constraints affected the quality of life.

of Systematic Reviews database (PROSPERO) registration number (CRD42021269897).

**Funding:** The authors received no specific funding for this work.

**Competing interests:** The authors have declared that no competing interests exist.

## Background

The World Health Organization (WHO) declared coronavirus disease of 2019 (COVID-19) an epidemic and public health emergency of international concern on January 30th, 2020. The virus is known to have originated from Wuhan City, Hubei Province, China in December 2019. COVID-19 drew global attention due to rapid increase in the numbers reported both in China and internationally within shortest period [1]. By February 20th, 2020, the number of contaminated COVID-19 cases in China reached a cumulative total of 75,465 cases and it had already spread to more than 25 countries among them Germany, Italy, France, Japan, Malaysia, Singapore, South Korea, Spain, Thailand, Vietnam, the United Arab Emirates, the United Kingdom (UK), the United States of America (USA) and Africa [2]. According to WHO (2021), a total of 190,597,409 confirmed cases of COVID-19, among them 4,093,145 deaths and only 3,430,051,539 vaccine doses have already been administered worldwide by 18th July 2021. Psychological conditions such as depression, anxiety and stress can contribute to the deterioration of quality of life (QoL) of populations. A Spanish study assessed the impact of COVID-19 on mental health and find that the most prevalent mental disorders were anxiety, sleep and affective disorders as well as depression with a considerable increase in suicidal behavior among women and men over 70 years old [3]. A national study in France reported a burnout of 55% during COVID-19 pandemic and he also find out that there was a strong link between the severity of the burnout syndrome, QoL and the impact of COVID-19 pandemic [2, 4]. Health related quality of life (HRQoL) is considered multidimensional and subjective and is assessed by patients using patient reported outcome measures (PROMs). According to WHO, HRQoL is defined as the general perception of individuals of their position in life (i) considering, the culture and value systems and (ii) in relation to expectations, goals, standards, and concerns [4]. HRQoL considers a wide-ranging concept influenced in a complex and interconnected manner by the psychological state, physical health, personal beliefs, social relationships and relationship to prominent features of the environment [5]. A systematic review discussed the impact of COVID-19 on the HRQoL on children and adolescents. Their results showed that lockdown significantly affected QoL, happiness and optimism (p < 0.001), as well as perceived stress. In their findings, the authors reported that only 15.3% (n = 146) of children and adolescents had low QoL before COVID-19 outbreak and during the pandemic, 40.2% of them reported low QoL [6]. A study conducted in the Kingdom of Saudi Arabia [7] assessed the QoL during COVID-19 in the general population and reported that being male (OR = 1.96; 95% CI = [1.31–2.94]), aged between 26 to 35 years (OR = 5.1; 95% CI = [1.33–19.37]), non-Saudi participants (OR = 1.69; 95% CI = [1.06–2.57]), individuals with chronic diseases (OR = 2.15; 95% CI = [1.33–3.48]), loss of job (OR = 2.18; 95% CI = [1.04–4.57]) and participants with depression (OR = 5.70; 95% CI = [3.59–9.05]), anxiety (OR = 5.47; 95% CI = [3.38–8.84]) and stress (OR = 6.55; 95% CI = [4.01–10.70]) were at a high risk of having lower levels of QoL during COVID-19 pandemic and lockdown period [7]. Swedish authors assessed the changes of QoL of the Swedish population using data of February and April 2020 and reported that on visual analogue scale (VAS), the mean QoL reduced from 77.1(SD:17.7) in February to 68.7(SD:68.7) in April 2020, a reduction of 8.4% pre and post pandemic measurements (P<0.000) [8]. In 2021, authors compared the QoL of Brazilian dietitians before (3.83 ± 0.59) and during COVID-19 pandemic (3.36 ± 0.66) and find that the results were statistically different [9]. To the best of our knowledge, this is one of the first systematic reviews to assess the impact of COVID-19 on HRQoL in the general population. The aim and objective of this systematic review is to assess the impact of COVID-19 and associated factors to Health Related Quality of Life in the general population.

## Methods

### Design and protocol

This systematic review was conducted and reported according to the Preferred Reporting Items for Systematic Reviews and Meta-Analyses (PRISMA)(Fig 1) [10]. A protocol was registered in the international Prospective Register of Systematic Reviews database (PROSPERO) with the registration number CRD42021269897.

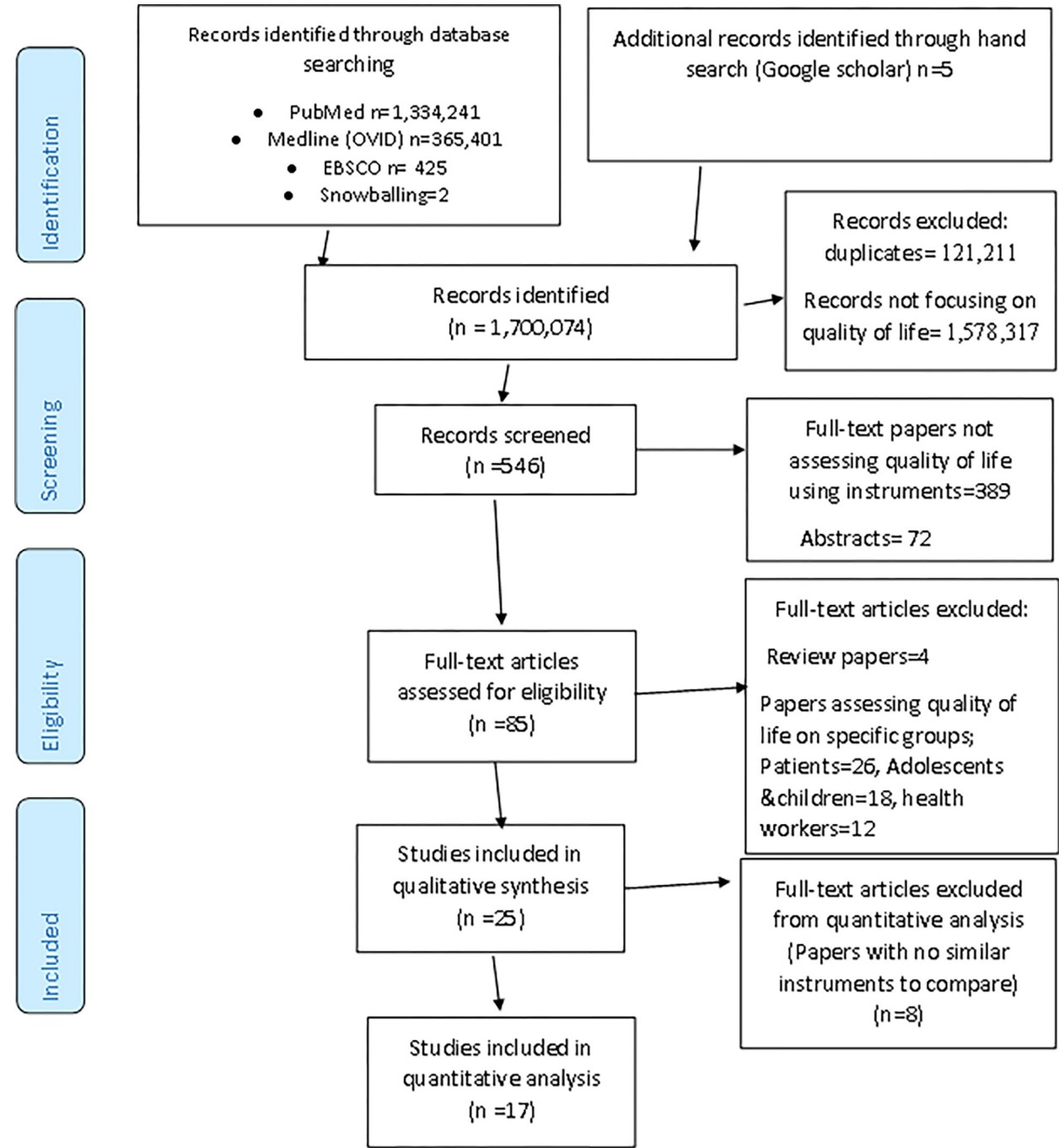

**Fig 1. Preferred Reporting Items for Systematic Reviews and Meta Analyses (PRISMA) flow diagram.**

**Table 1. Inclusion and exclusion criteria.**

| Inclusion | Exclusion |
|---|---|
| Primary, empirical, quantitative, cross-sectional, cohort, case control, peer reviewed, assessing effects of COVID-19 on quality of life before and during COVID in general population, used a validated instrument, is in English language | Articles not assessing quality of life, focusing on subgroups of populations such as health workers, population with previous mental health, population with cancer, HIV or any other chronic disease, secondary and non-empirical, non-peer reviews, review articles such as scoping, narrative and Systematic reviews, papers on Medrxiv and SSRN server, comments, letters, conference abstracts, books and book chapters, articles not assessing quality of life, papers with previous mental health problems or papers not assessing quality of life in the general population during the COVID-19 pandemic |

## Eligibility

Articles were included if they were (i) primary and empirical, quantitative, cross-sectional, cohort, case-control, peer reviewed, assessing effects of COVID-19 on the quality of life during COVID-19 in the general population, utilized validated scales for measurement, published in English language from inception to June 30[th], 2022. Articles were excluded if (ii) focusing on subgroups of populations such as health care workers, population with previous mental health, population with cancer, HIV or any other chronic disease, utilized secondary data and non-empirical, non-peer review, review articles such as scoping, narrative or Systematic reviews, papers on Medrxiv and SSRN server, comments, letters, conference abstracts, books and book chapters, articles not assessing the quality of life, papers on the population with previous mental health or papers not assessing quality of life in the general population during COVID-19 pandemic. There was no limit on the number of papers to synthetize. All articles satisfied the eligibility criteria were included. Grey literature was used only to support the background section of the research (Table 1).

## Search strategy and selection

A comprehensive electronic search in the PubMed, EBSCO Host Research Databases (Academic Research Complete), MEDLINE (OVID) and Google scholar search engine was conducted from January 5[th], 2022 to February 28[th], 2022 and updated on June 30[th], 2022. The search strategy and data extraction were designed by DAN, DK and JG using Medical Subject Headings (MeSH), field tags and relevant keywords related to quality of life, COVID-19 and general population. Boolean operators, thesaurus, truncation, nesting and quotation marks were used to strengthen the search. The full search strategy was provided in supplementary documents. Additional search of the references from retrieved systematic reviews through snow balling was performed. All retrieved papers were downloaded and saved to Mendeley for intext citations and referencing. The following was used as search string for PubMed; "("Quality of Life"[Mesh] OR "quality of life" [tw] OR "Health-related Quality of Life" [tw] OR HRQoL[tw]) AND ("COVID-19"[Mesh] OR COVID-19[tw] OR "SARS-CoV-2" [tw] OR Sars-cov-2[tw] OR Coronavirus[tw] OR SARS OR "Coronavirus disease 2019" [tw] OR "severe acute respiratory syndrome coronavirus 2" [tw] OR "2019-nCoV Infection" [tw] OR 2019-nCoV[tw] OR "COVID-19 Virus Disease" [tw]) AND (Population[Mesh] OR "general population"[tw] OR "general public"[tw] OR public[tw] OR communit*[tw]).

## Data extraction technique

A standardized data collection tool to extract relevant information from papers was designed. The following data was collected; authors, country of publication, study design, sample size,

demographic characteristics, HRQoL before COVID-19, HRQoL during COVID-19, QoL measurement tool, statistical tests and risk factors as well as their odds ratios (OR). Data was extracted by two authors (DAN & BW) and verified by the second author (JNM). Discrepancies were resolved by the 3rd author (JG).

## Quality appraisal

Two authors (DAN and JG) independently assessed the quality of the included papers using a modified Newcastle-Ottawa Scale (NOS) modified for cross-sectional studies. The quality criteria used in cross-sectional studies were: sample representation, sample size, response rate and validated measurement tools with appropriate cut-off points and the control of confounding variables or use of multiple regression. The quality score ranged between 0 and 5 and any study scoring > or = 3 was considered as high and any study scoring < 3 was considered to be at low quality.

## Main outcome

Health related quality of life (HRQoL).

## Measures of effect

Health related quality of life measurements such as means of EQ5D and WHOQoL-BREF and their standard deviations were calculated.

## Heterogeneity and risk of bias assessment of included studies

The risk of bias, heterogeneity and mean effect size were assessed for EQ5D and WHOQoL_-BREF. A random-effect model was fitted to the data. The amount of heterogeneity (i.e., $\tau^2$), was estimated using the restricted maximum-likelihood estimator [11]. In addition to the estimate of $\tau^2$, the $Q$-test for heterogeneity [12] and the $I^2$ statistic [13] are reported. In case any amount of heterogeneity is detected (i.e., $\hat{\tau}^2 > 0$, regardless of the results of the $Q$-test), a prediction interval for the true outcomes is also provided [14]. Studentized residuals and Cook's distances are used to examine whether studies may be outliers and/or influential in the context of the model [15]. Studies with a studentized residual larger than the $100\times(1-0.05/(2\times k))$th percentile of a standard normal distribution are considered potential outliers (i.e., using a Bonferroni correction with two-sided $\alpha = 0.05$ for $k$ studies included in the meta-analysis). Studies with a Cook's distance larger than the median plus six times the interquartile range of the Cook's distances are considered to be influential. The rank correlation test [16] and the regression test [17], using the standard error of the observed outcomes as predictor, are used to check for funnel plot asymmetry. The analysis was carried out using R (version 4.2.1) (R Core Team, 2020) and the **metafor** package (version 3.8.1) [18].

## Qualitative synthesis and quantitative analysis

Data was summarized following the "Institute of Medicine committee on the standards for systematic reviews of comparative effectiveness research: Finding out what works in health care; standards for systematic reviews: recommended standards for qualitative synthesis" [19] and the key characteristics of included studies if similar were grouped, synthetized qualitatively and discussed in order to draw conclusions. The mean effect size was performed and pooled for both EQ5D and WHOQoL_BREF using Random effect model. In meta-analysis, they are two classes of models; fixed and random effect models. For fixed-effect model, all studies are assumed that population effect sizes are the same and are appropriate for drawing inferences

on the studies included in the meta-analysis whereas random-effect model attempt to generalize the findings beyond included studies and assume that the selected studies are random samples from a larger population. According to Dettori et al. (2022), the observed effect size is a combination of the study-specific effect and the sampling error [20]. The model is: Yi = B random+Ui+ei, where B random is the average of the true effect sizes, Ui addition of random effect, ei = error. Homogeneity of effect sizes, that is $\tau 2 = 0$ can be tested by chi-square statistic which is Q statistic. The $\tau 2$ can be used to estimate the degree of heterogeneity. $\tau 2$ also depends on the type of effect size used and the common one is I2. I2 is interpreted as the proportion of between-study heterogeneity to the total variation (between–study heterogeneity plus sampling error). When I2 is negative, it is truncated to zero. I2 of 25, 50 and 75% is considered low, moderate and high heterogeneity respectively as a rule of thumb [21]. When conducting a random effect model, it is required to estimate the amount of heterogeneity. The most widely used heterogeneity estimator in medical science is DerSimonian and Laid. Other estimators such as maximum likelihood and restricted maximum likelihood may also be used.

## Results

A total number of 1,700,074 articles were identified from electronic databases on PubMed (1,334,241), Medline (OVID) (365,401), EBSCO (Host Research Databases (Academic Research Complete)) (425) and manual search with Google scholar search engine (5). 121,211 duplicates and 1,578,317 papers not related to quality of life were removed and 546 papers were retained. 461 full text papers and abstracts were removed to retain 85 full articles for screening. Finally, 25 full text articles were included for quality synthesis. 8 full articles were excluded because there were no papers with similar instruments to compare and 17 studies were included for quantitative analysis (PRISMA) (Fig 1).

### Characteristics of included studies

The total sample size of included studies was N = 22,967 participants and ranges from 225 to 3,002 participants per study. The majority (64.85%) were female (n = 14,894). 3 studies were done in China [22–24], 2 Morocco [25, 26], 2 Vietnam [27, 28], 2 Italy [29, 30], 1 Saudi Arabia [7], 1 Malaysia [31, 32], 1 Jordan [33], 1 Philippines [34], 1 Hong Kong [35], 1 Portugal [36], 1 Israel [37], 1 Spain [38], 1 Brazil [39], 1 Scotland [40], 1 USA [41], 1 Egypt [42], 1 study done in two countries Belgium and Netherlands [43] and one in Africa, North America, Asia, Australia, Europe, South America [44]. Nine articles used the WHOQoL-BREF tool to measure the quality of life in their respective countries [7, 30–33, 35, 37, 39, 41], six papers utilized the EQ-5D [22, 25, 27, 28, 36, 43], three used SF12/SF-8/ SF36 [24, 26, 38], one utilized EQLS [40], one used GH12 [29], one utilized the COVID-19 QoL questionnaire [44], on used the COVID-19 (COV19- Impact on the quality of life (COV19-QoL) scale) [42] and one utilized MLT [34]. The majority of the studies (n = 23) were of cross-sectional design and only one [40] was of a mixed method. Nine studies utilized the WHOQoL-BREF [7, 30–33, 35, 37, 39, 41], nine utilized the EQ-5D [22, 25, 27, 28, 36, 40, 43] among them two studies compared and reported QoL scores before and during COVID-19 [25, 36], three articles utilized the SF12/SF-8/SF36 [24, 26, 38] one article utilized EQLS [40], one paper used GH12 [29]. Another one assessed the HRQoL using COVID-19 QoL questionnaire [44] and one article used MLT [34] to assess HRQoL in general population (Table 2).

### Measurement tools

The most used instruments in this study (WHOoQoL and EQ5D) are explained below and a brief description of their normal values for unaffected populations are given at the beginning

**Table 2. Mean scores of QoL of included studies.**

| Study | Measurement method | QoL before COVID-19 | Mean QoL during COVID-19 | Most affected domain | Least affected domain |
|---|---|---|---|---|---|
| 1 Algahtani et al. 2021 [7] | WHOQOL-BREF | NR | 39% (SD = NA) | Social-relationship | Environment |
| 2 Abdullah et al. 2021 [31] | WHOQOL-BREF | NR | 69.44% (SD = 12.78) | Social relationship | Environment |
| 3. Al-Shannaq et al 2021 [33] | WHOQOL-BREF | NR | 73.21% (SD ¼ 16.17) | Social relationship | Environment |
| 4 Aruta et al. 2022 [34] | MLT | NR | NA | NA | NA |
| 5 Azizi et al 2020 [25] | EQ-5D-5L | 0.91 (SD: NR) | During confinement (0.86; P<0.0001 Before confinement (utility = 0.91) | Mobility | Anxiety/depression |
| 6 Ballegooijen et al 2021 [43] | EQ-5D | NR | 0.79 (0.77–0.81) Belgium 0.84 (0.82–0.86) Netherlands | NA | NA |
| 7 Bonichini & Tremolada, 2021 [29] | GH12 | NR | 17.86 (SD 5.85) | NA | NA |
| 8 Chen et al. 2021 [22] | EQ5D | NR | 0.990 (SD:N/A) | NA | NA |
| 9 Choi et al 2021 [35] | WHOQoL | NR | 61.685% (SD = 14.6275) | Physical health | Environmental health |
| 10 Epifanio et al. 2021 [30] | WHOQoL | NR | 54.48% (SD = 7.77) | Physical domain | Environment |
| 11 Ferreira et al. 2021 [36] | EQ-5D-5L | 0.887 (SD = 0.005) | 0.861 (SD = 0.027) (During COVID_19) 0.887 (SD = 0.005) (pre-COVID_19) | Anxiety/depression | Self-care |
| 12 Khodami et al. 2022 [44] | COVID-19 QoL questionnaire | NR | 21(SD = 21.18) (5(SD: 4.789)) | NA | NA |
| 13 Lipskaya-Velikovsky (2021) [37] | WHOQoL | NR | 73.5% (IQR:59.5–87.5) SD: NA | Physical | Environment |
| 14 Iglesias-López et al 2021 [38] | SF-36 (SF-36v2) | NR | NA | Physical functioning | Mental health |
| 15. Ping et al. 2020 [23] | EQ-5D | NR | 0.949 (SD: 0.102) 85.52(SD: 19.37) VAS | Pain/disorder | Anxiety/depression |
| 16 Qi et al. 2020 [24] | SF8 | NR | 75.3 (SD = 16.6) | physical | mental |
| 17 Tran et al. 2020 [27] | EQ-5D-5L and EQ-VAS | NR | 0.95 (± 0.07) and 88.2 (SD: ± 11.0) | Anxiety/depression | Self-care |
| 18 Vitorino et al. 2021 [39] | WHOQoL | NR | 14.325 (SD = 2.8625) | Physical | anxiety |
| 19 Quynh Vu et al. 2020 [28] | EQ-5D-5L | NR | 0.95 (general pop) | NA | NA |
| 20 Yee et al. 2021 [32] | WHOQOL-BREF | NR | 13.2 (SD = 2.8325) | Social relationship | Environment |
| 21 Campbell & Davison,2022 [40] | EQLS | NR | NA | NA | NA |
| 22 Hansel et al 2022 [41] | WHOQOL-BREF | NR | 25.1 (SD = 4.9) | NA | NA |
| 23 Mohsen et al. 2022 [42] | COV19-Questionnaire_Quality of Life | NR | 2.3 (SD = ±0.6) | Quality of life in general and perception of danger | Perception of mental health perception |
| 24 Samlani Z1 et al. 2020 [26] | SF12 | NR | 70.60 (SD: ±13.1) | Mental health | Mental health with chronic disease |
| 25. Teotônio et al 2020 [50] | WHOQoL-BREEF | NR | 62.59 (SD = 11.54) | Economic | Physical |

EQLS: European Quality of Life Survey, EQ-5D/EQ-5D-5L: EuroQoL-Five dimensions, WHO_OoL-BREF: World Health Organization Quality of Life BREF; SF12: Short form; COV-19 QoL questionnaire; GH12: The General Health Questionnaire; MLT: My Life Today Questionnaire

of each reported instrument. Eight (n = 8) scales have been used to assess health related quality of life on the general population worldwide during COVID-19. EQ-5D: Euro_QoL-Five dimensions; is a preference and generic quality of life instrument to valuate and describe health related quality of life; the higher the index, the better the health. It describes health in terms of five dimensions; mobility, self-care, usual activities, pain/discomfort and anxiety/depression [45]. A utility score can be generated from the five dimensions based on a published algorithm with a value of 0 for death and 1 for perfect health. WHO_OoL-BREF: the WHO_BREF is a 26-item instrument with four domains: physical health (7 items), psychological health (6 items), social relationships (3 items), and environmental health (8 items) [46]. It is scored from 1 to 5 on a response scale but transformed linearly to a 0 to 100 scale. 0 point represent the worse possible health state while 100 points represent the best possible health state. SF12: Short form are generic health survey short-forms (don't use preference based approach) to assess quality of life which are used in research and clinical practice, health policy and general surveys [47]. EQLS: European Quality of Life Survey is a 2012 scale which considers the following dimensions; employment and work-life balance, family and social life, health and public services, home and local environment, quality of society, social exclusion and community involvement, standard of living and deprivation, subjective well-being which is designed for the general population[48]. GHQ: General Health Questionnaire is a measure of current mental health and since its development by Goldberg in the 1970s it has been extensively used in different settings and different cultures [49]. COV19-QoL is a 6-item scale covering main areas of quality of life in relation to mental health. The first item covers patients' feelings about the impact of the current pandemic on their quality of life in general population. The second and third include the participants' perceptions of possible mental and physical health deterioration. COV19-QoL scale is a recently developed specific reliable and valid tool assessing perceptions of deterioration in QoL as a result of the COVID-19 pandemic [42]. MLT: My Life Today the 9-tem (4) scale was used to measure the participants' perceptions of various life domains, including the assessment of life in general population [34].

## Quality of life before and during COVID-19

Among 25 articles reporting changes in QoL, 23 reported the mean QoL only during COVID-19 and did not report the QoL before COVID-19. Nine papers [22, 25, 27, 28, 36, 40, 43] utilized EQ5D among them only two reported both QoL before COVID-19 as compared to that of during COVID-19 [25, 36] using EQ5D instrument. Azizi et al. (2020) in Morocco reported an EQ5D mean score before COVID-19 of 0.91(SD: NR) and 0.86 (SD: NR) during the pandemic. This makes a drop of 0.05 on QoL. Ferreira et al. (2021) in Portugal also reported an EQ5D mean score before COVID-19 of 0.887 (SD: NR) and 0.861 (SD: NR) during COVID-19 making a drop of 0.026 of QoL. Using EQ5D, the minimum score reported during COVID-19 was 0.79 (SD: 0.17–1.41) and a maximum of 0.95 (SD:.14–1.76) with a mean score 0.89 (SD: 0.66–1.13). Among papers reporting QoL using WHOQoL, no study reported both scores (before and after) and the WHOQoL minimum score reported during COVID-19 was 13.20 (SD: 9.85, 16.55) with a maximum of 73.50 (SD:66.14, 80.86). The mean reported was 53.38 (SD:38.50, 68.27). The lower the score, the lower the QoL. The rest of quality of life instruments were used at least once making it not practical to report their means for a comparison.

## Forest plot WHO_BREF

A total of $k$ = 9 studies were included in the analysis. The observed outcomes ranged from 13.2000 to 73.5000, with the majority of estimates being positive (100%). The estimated mean outcome based on the random-effects model was $\hat{\mu} = 53.3841$ (95% CI: 38.4992 to 68.2691).

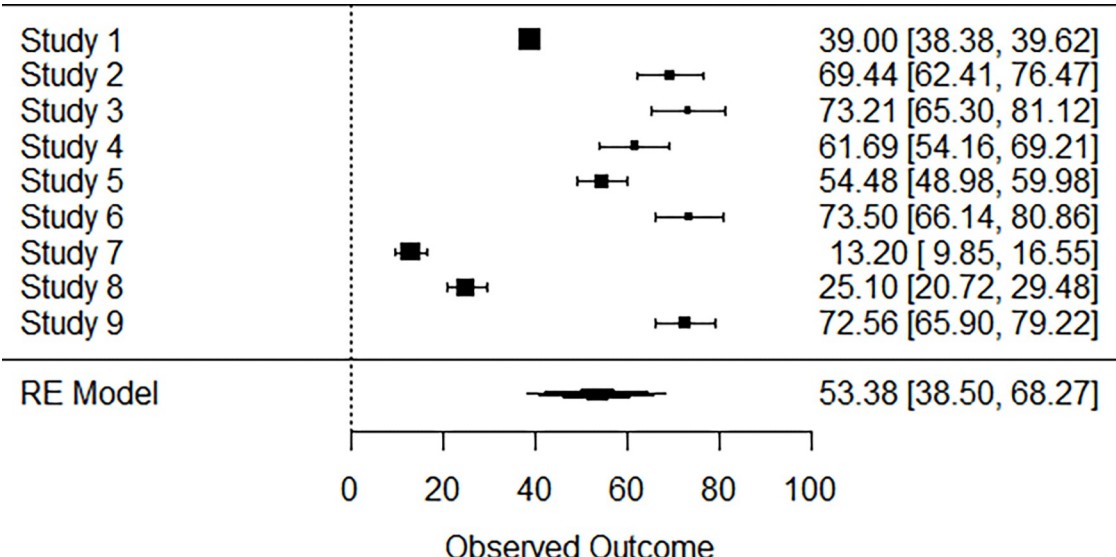

**Fig 2. Mean score using WHOQoL-BREF forest plot.** Study1 [7], study2 [31], Study3 [33], Study4 [35], Study5 [7], Study6 [37], Study 7 [32], Study8 [41], study 9 [50]. The mean health related quality of life using WHOQoL_BREF is estimated at 53.38.

Therefore, the average outcome differed significantly from zero ($z = 7.0293$, $p<0.0001$). A forest plot showing the observed outcomes and the estimate based on the random-effects model is shown (Fig 2)

## Forest plot EQ5D

A total of $k = 8$ studies were included in the analysis. The observed outcomes ranged from 0.7900 to 0.9900, with the majority of estimates being positive (100%). The estimated average outcome based on the random-effects model was $\hat{\mu} = 0.8930$ (95% CI: 0.6566 to 1.1295). Therefore, the average outcome differed significantly from zero ($z = 7.4024$, $p<0.0001$). A forest plot showing the observed outcomes and the estimate based on the random-effect model is shown (Fig 3)

## Heterogeneity and risk of bias of WHO-BREF studies

According to the $Q$-test, the true outcomes appear to be heterogeneous ($Q(8) = 656.8283$, $p<0.0001$, $\hat{\tau}^2 = 509.6543$, $I^2 = 99.2019\%$). A 95% prediction interval for the true outcome is given by 6.7003 to 100.0680. Hence, even though there may be some heterogeneity, the true outcomes of the studies are generally in the same direction as the estimated average outcome.

A funnel plot of the estimates is shown in (Fig 4). The regression test indicated funnel plot asymmetry ($p = 0.0019$) but not the rank correlation test ($p = 0.1194$) (Fig 4).

## Heterogeneity and risk of bias of EQ5D studies

According to the $Q$-test, there was no significant amount of heterogeneity in the true outcomes ($Q(7) = 0.3149$, $p = 0.9999$, $\hat{\tau}^2 = 0.0000$, $I^2 = 0.0000\%$). A funnel plot of the estimates is shown in (Fig 5). Neither the rank correlation nor the regression test indicated any funnel plot asymmetry ($p = 0.5664$ and $p = 0.8657$, respectively) (Fig 5).

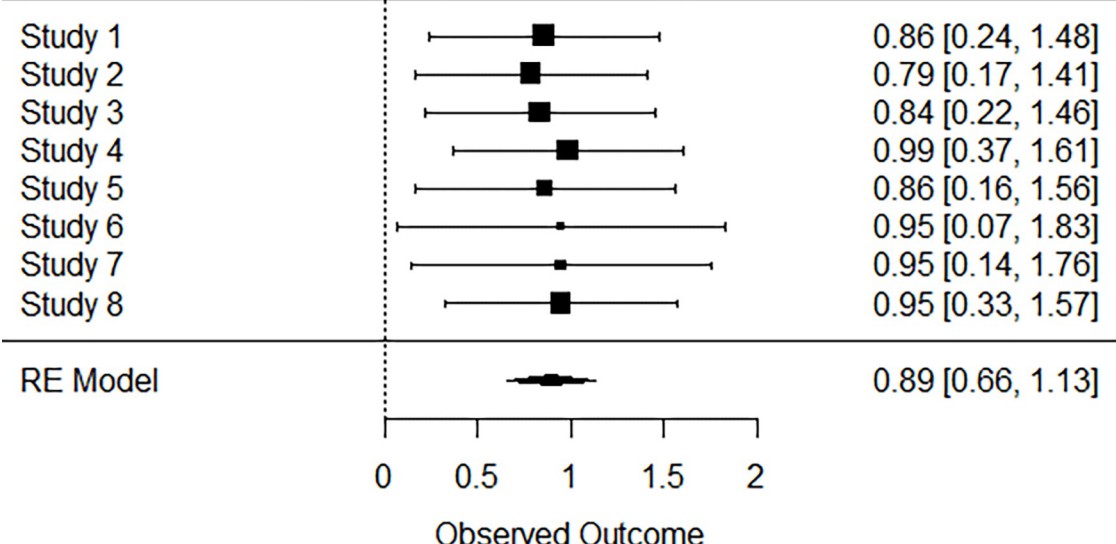

**Fig 3. Mean score using EQ5D forest plot.** Study 1 [25], Study2 [43], Study3 [43], Study4 [22], Study5 [36], Study6 [23], Study7 [27], Study8 [28].

## Quality assessment

We used the Modified Newcastle-Ottawa quality assessment scale tool to assess the quality of included papers and only 3 papers scored five out five (5/5) [29, 36, 39]. Ten papers scored four out of five (4/5) [7, 22, 23, 30, 31, 33, 35, 38, 42, 44]. Eight studies scored three out five (3/5) [24, 27, 28, 32, 34, 38, 43]. Only three papers [25, 26, 40] scored 2 out five making it the lowest score and therefore low quality. We included the three low quality articles in the qualitative

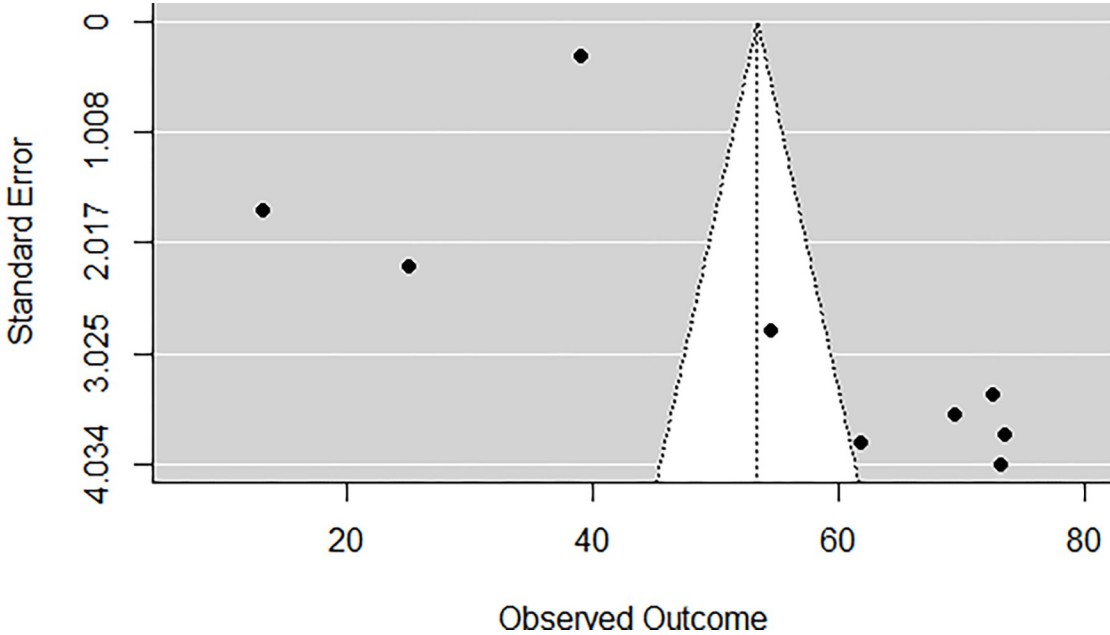

**Fig 4. WHOQoL-BREF funnel plot.** Study1 [7], study2 [31], Study3 [33], Study4 [35], Study5 [7], Study6 [37], Study 7 [32], Study8 [41], study 9 [50].

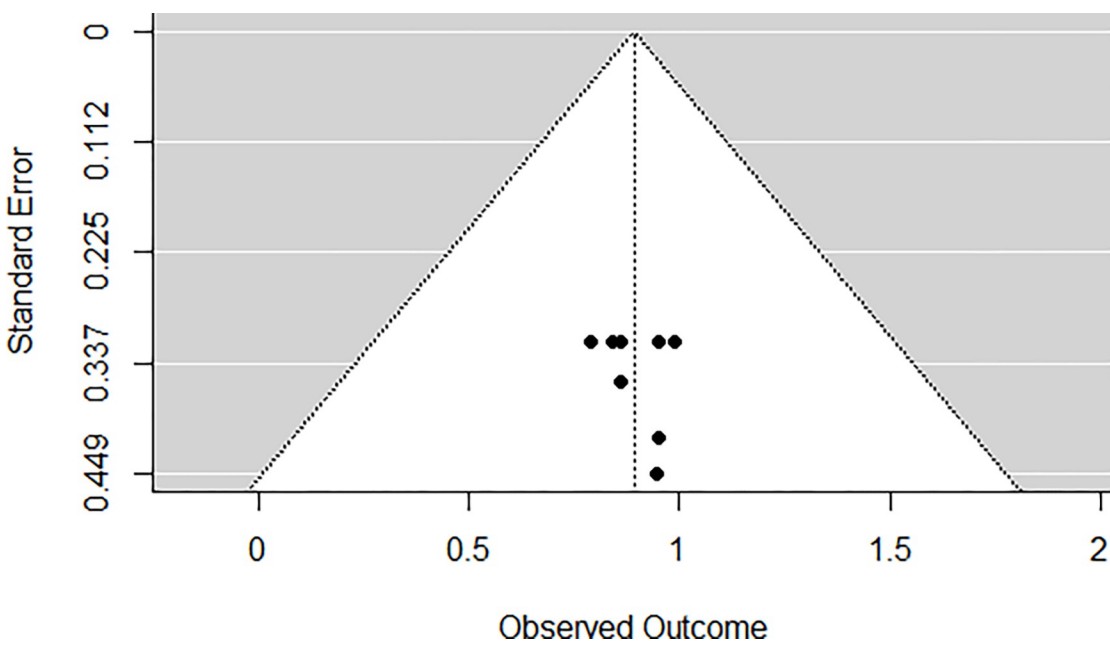

**Fig 5. EQ5D funnel plot.** Study 1 [25], Study2 [43], Study3 [43], Study4 [22], Study5 [36], Study6 [23], Study7 [27], Study8 [28].

synthesis however only one low quality paper [25] was included in the quantitative analysis (meta-analysis) because it was lying within two standard deviations of the mean therefore it was not affecting the results (Table 3).

## Quality of life and factors associated to the low HRQoL

**EQ-5D.** Eight studies [22, 25, 27, 28, 36, 40, 43] have used the EQ-5D to assess the quality of life for the general population during COVID-19 pandemic. The mean score using EQ-5D estimated at 0.89 [95% CI 0.66–1.13]. By using EQ-5D, authors [25] reported the results (before and during confinement) that the quality of life was affected in the five health dimensions; mobility 87%(87%), self-care 97%(93%), usual activities 82%(89%), pain/discomfort 70%(78%) and anxiety/depression 44%(66%). His comparison on the two samples showed that during confinement, peoples had lower scores of HRQoL at 0.86 (p<0.001) as compared to before confinement whose score was 0.91 [25]. Female gender was affected with lower scores of HRQoL than their counterpart male on both utility (0.85; P = <0.0001 and VAS (78.49; P = 0.004) and (utility = 0.89 and VAS = 83.78) respectively. Marital status was significantly associated to EQ-5D utility (P = 0.002) and VAS (P = 0.005) scores, widowed had the worst HRQoL (utility = 0.43 and VAS = 48.75) compared to single (utility = 0.87 and VAS = 80.09), married (utility = 0.86 and VAS = 81.43), and separated (utility = 0.89 and VAS = 80.15) participants. Participants with university education had the higher EQ-5D utility score (0.88; p<0.001) and age did not have a significant impact. A study done in Belgium and Netherlands also evaluated the quality of life using EQ5D as well, a minority in both countries felt stressed with 27% and 14% respectively [43]. The majority reported concerns about their personal current and future financial situation (59 and 48% respectively) and the national economies (88 and 86%). Specifically, in Belgium, the EQ-5D before COVID-19 measured 0.82 (95% CI; 0.80–0.84) and during COVID-19 measures 0.79 (95% CI; 0.77–0.81). In Netherlands, before COVID-19, 0.85 (95% CI; 0.83–87) and during COVID-19 outbreak, it was 0.84 (95% CI; 0.82–0.86). Chen et al. (2021), using EQ5D concluded that the mean EQ-5D score and VAS

**Table 3. Modified Newcastle-Ottawa quality assessment scale.**

| Author & year of publication | Sample representativeness | Sample size (>500) | Response rate | Study used validate measurement tools with appropriate cut-offs | Control of confounding variables or used multiple regression | Score |
|---|---|---|---|---|---|---|
| 1.Algahtani et al. 2021 | + | + | - | + | + | 4 |
| 2. Abdullah et al. 2021 | + | - | + | + | + | 4 |
| 3. Al-Shannaq et al 2021 | + | - | + | + | + | 4 |
| 4. Aruta et al. 2022 | + | - | _ | + | + | 3 |
| 5. Azizi et al 2020 | + | - | - | + | - | 2 |
| 6. Ballegooijen et al 2021 | + | + | - | + | - | 3 |
| 7. Bonichini & Tremolada, 2021 | + | + | + | + | + | 5 |
| 8. Chen et al. 2021 | + | + | - | + | + | 4 |
| 9. Choi et al 2021 | + | - | + | + | + | 4 |
| 10. Epifanio et al. 2021 | + | + | _ | + | + | 4 |
| 11. Ferreira et al. 2021 | + | + | + | + | + | 5 |
| 12. Khodami et al. 2022 | + | + | - | + | + | 4 |
| 13. Lipskaya-Velikovsky (2021) | + | - | _ | + | + | 3 |
| 14. Iglesias-López et al 2021 | + | | + | + | + | 4 |
| 15. Ping et al. 2020 | + | + | _ | + | + | 4 |
| 16. Qi et al. 2020 | + | + | - | + | _ | 3 |
| 17 Tran et al. 2020 | + | - | _ | + | + | 3 |
| 18. Vitorino et al. 2021 | + | + | + | + | + | 5 |
| 19. Vu et al. 2020 | + | - | + | + | - | 3 |
| 20. Yee et al. 2021 | + | - | - | + | + | 3 |
| 21. Campbell and Davison,2022 | + | - | - | + | _ | 2 |
| 22. Tonya Cross Hansel et al 2022 | + | | _ | + | + | 3 |
| 23. Shorouk Mohsen et al. 2022 | + | - | _ | + | + | 4 |
| 24. Samlani Z1 et al. 2020 | + | - | _ | + | - | 2 |
| 25. Teotônio et al 2020 | + | + | + | + | - | 4 |

Key: 1. Sample representativeness (= 65% of the sample), 2.sample size> 600 participants, 3. response rate>80%, 4. study used validate measurement tools with appropriate cut-offs and 5 control of confounding variables or used multiple regression

were 0.99 and 93.5. Their multiple linear regression showed that the quality of life measure was related to physical activities ($\beta$ = 0.006) and keeping home ventilation ($\beta$ = 0.063) in Daqing, and were related to wearing a mask when going out ($\beta$ = 0.014), keeping home ventilation ($\beta$ = 0.061), other marital status ($\beta$ = − 0.011), worry about the epidemic ($\beta$ = − 0.005) and having a centralized or home quarantine ($\beta$ = − 0.005) in Taizhou [22]. Using EQ5D, authors concluded that those quarantined at home experienced higher levels of anxiety and a lower HRQoL compared with the pre-COVID-19 pandemic population. Females and elderly individuals experienced the highest levels of anxiety and poorest HRQoL (OR not reported) [36]. Other authors [23] using the same instrument EQ5D reported that the risk of pain/discomfort and anxiety/depression in general population in China raised significantly with aging, with

chronic disease, lower income, epidemic effects, worried about get COVID-19 during the COVID-19 pandemic (OR not reported) [23]. Tran et al. (2020) With the same instrument EQ5D (n = 341) reported that 66.9% of household income loss was due to the impact of COVID-19. The mean score of EQ-5D and EQ-VAS was 0.95 (SD ± 0.07) and 88.2 (SD ± 11.0) respectively. The domain of Anxiety/Depression had the highest proportion of reporting any problems among 5 dimensions of EQ-5D (38.7%). Being female, having chronic conditions and living in the family with 3–5 members were associated with lower HRQOL score (OR not reported) [27]. Vu et al. (2020) using EQ5D reported the highest mean EQ-VAS at 90.5 (SD: 7.98) among people in government quarantine facilities, followed by 88.54 (SD: 12.24) among general population and 86.54 (SD 13.69) among people in self-isolation group [28]. The EQ-5D value was reported as the highest among general population at 0.95 (SD: 0.07), followed by 0.94 (SD: 0.12) among people in government quarantine facilities, and 0.93 (SD: 0.13) among people who put themselves in self-isolation. Overall, most people, at any level, reported having problems with anxiety and/or depression in all groups.

**WHOQoL-BREF.**   The WHOQoL average scores was estimated at 50.55% 95% CI [32.19–68.90]. Authors by using WHOQoL-BREF reported a quality of life affected with a score of 39% (CI not reported) and according to authors, males were more affected with OR = 1.96 (95% CI = 1.31–2.94); participants aged 26 to 35 years OR = 5.1; (95% CI = 1.33–19.37); non-Saudi participants OR = 1.69 (95% CI = 1.06–2.57); individuals with chronic diseases OR = 2.15 (95% CI = 1.33–3.48); those who lost their job OR = 2.18 (95% CI = 1.04–4.57) and those with depression OR = 5.70 (95% CI = 3.59–9.05), anxiety OR = 5.47; (95% CI = 3.38–8.84), and stress OR = 6.55 (95% CI = 4.01–10.70) [7].

In 2021, a study [31] concluded that higher psychological QoL reduced the odds of depressive symptoms OR = 0.83 (95% CI = 0.69–0.99, p = 0.032) and depressive with comorbid anxiety symptoms OR = 0.82, (95% CI = 0.68–0.98, p = 0.041), whereas higher physical health QoL OR = 0.85, (95% CI = 0.75–0.97, p = 0.021) and social relationship QoL OR = 0.70 (95% CI = 0.55–0.90, p = 0.009) reduced the odds of anxiety symptoms [31]. In 2020, a study [33] had reported a mean for total QoL score of 73.21 (SD ¼ 16.17). The mean general QoL and health scores were 3.15 (SD ¼ 0.94) and 3.40 (SD ¼ 0.95). As for the four QoL subscales, the mean scores in each domain were as follows: 18.04 (SD ¼ 4.39) for physical health, 17.65 (SD ¼ 3.77) for psychological health, 8.69 (SD ¼ 2.67) for social relationships, and 22.29 (SD ¼ 5.84) for environment(29). Choi et al (2021), using the same QoL scale reported that 69.6% of participants were worried about contracting COVID-19, and 41.4% frequently suspected themselves of being infected whereas 29.0% were concerned by the lack of disinfectants. All of these findings were associated with poorer HRQoL in the physical and psychological health, social relationships, and environment domains (OR not reported). 47.4% of participants were concerned that they may lose their job because of the pandemic and 39.4% were bothered by the insufficient supply of surgical masks [35]. The results of a study [30] showed statistically significant difference in QoL depending on a number of variables, including sex, area of residence in Italy, and being diagnosed with a medical/psychiatric condition (OR: NR). The overall average score at the WHOQoL-BREF was 54.48 (SD = 7.77). The item with the lowest scores was 14 (about the use of spare time), given that 932 (41.4%) participants reported to have little or no time for leisure at the time of data collection. Regarding the other three domains of the WHOQoL, items with lowest scores were: item 15 for the physical domain, as 1019 (45.3%) participants reported little or no possibility to do physical activity; item 5 for the psychological domain, with 712 (31.6%) respondents reporting that they were not enjoying their lives at the time of data collection, and item 21 for social relationships, as 843 (37.4%) respondents reported that they were little or not at all satisfied with their sexual life [30]. A research in 2021 [37] reported that COVID-19 has had a wide impact on the general

population, with the potential for negative secondary impacts. Women, young adults, and the unemployed are at high risk for secondary effects (ORs:NR). Another study [39] scores on the social relationships QoL domain were lower among participants who had a family member or friend with COVID-19 and among those who engaged in negative forms of spiritual religious coping (SRC). The quarantine during the COVID-19 pandemic has limited personal contact with family and friends, adversely affected sexual activity, and has restricted other activities that are assessed in the social relationships QoL domain. Positive forms of spiritual religious coping (SRC) were associated with better scores on this domain, as reported in other studies [37]. In 2020, a study [32] highlighted that approximately one in three individual experienced mild-to-severe depressions during the nationwide movement control order (MCO). The results of a study [41] reported that most would expect quality of life to be challenged during a global pandemic; however, when behavioral health assessed as a component of overall quality of life, longer term outcomes became concerning [41].

**SF12/SF-8/ SF36.**   Samlani *et al.* (2020) by using SF 12/8 (Chinese) scale, all participants obtained a total average score of 70.60 (±13.1) with a mental health score (MCS) of 34.49 (±6.44) and a physical health score (PCS) of 36.10 (± 5.82). The physical (PCS) and mental (MCS) scores of participants with chronic diseases were 32.51 (±7.14) and 29.28 (±1.23), respectively. Overall, the participants' PCS and MCS scores suffered from chronic diseases and the elderly participants were lower than those of young participants without comorbidities (23). López *et* al (2021) reported the following results using SF-36; the presence of pain in subjects undergoing confinement was persistent, with varying intensity and frequency based on age, gender, physical activity, and work status (OR:NR). In any of these conditions, the quality of life of the subjects in confinement has been severely affected [38]. Qi *et* al. 2020 using–SF8 (Chinese), participants' average physical component summary score (PCS) and mental component summary score (MCS) for HRQoL were 75.3 (SD = 16.6) and 66.6 (SD = 19.3), respectively. More than half of participants (53.0%) reported moderate levels of stress. Significant correlations between physical activity participation, QoL, and levels of perceived stress were observed ($p < 0.05$). Prolonged sitting time was also found to have a negative effect on QoL ($p < 0.05$) [24].

**EQLS.**   Campbell & Davison (2022) by using EQLS found that there are strong relationships between QoL and income, disability and living arrangement as well as social isolation and Disability and living arrangement [40]. Correlation and multiple regression analyses showed a strong relationship between social isolation, gratitude, uncertainty and QoL with social isolation being a significant predictor (OR not reported).

**GH12.**   Bonichini & Tremolada (2021) reported that the mean GH12 score in participants amounted to 17.86 (SD = 5.85), reflecting a contingent moderate stressful impact on QoL. GH12 identified 39% of respondents as having subclinical QoL scores (score $\geq$ 15). 24.5% of such respondents as having very problematic scores (score $\geq$ 19), and 36.5% of such respondents as having normal scores (score $<$ 15). Analysis of variance (ANOVA) showed there was a significant difference ($F(2, 1.836) = 5.50$, $p = 0.004$, $\eta 2 = 0.01$) in mean GH12 scores [29].

**COVID-19 QoL questionnaire.**   The results of Khodami et al. (2022) showed that Quality of life is significantly decreased over time, perceived stress level raised significantly and an increased level of difficulty in emotion regulation has happened. Younger peoples and individuals who had a worsening quality of life response tended to show more stress and emotion regulation problems [44]. Mohsen et al. (2022) using COVID-19 on Quality of life scale reported that the total COV19-QoL scale score was 2.3±0.6. Two items show the highest mean with 2.6 ±0.7 (quality of life in general and perception of danger on their personal safety) indicating the poorest quality of life regarding these 2 items. However, the lowest mean score was related to the perception of mental health deterioration (1.9±0.8). Significant variables in the bivariate

analysis revealed that sex (regression coefficient = 0.1 (95% CI(0.02 to 0.2), p value = 0.02), monthly income (regression coefficient (95% CI) = 0.1 (0.004 to 0.2), p value = 0.04), knowing someone infected with COVID19 (regression coefficient (95% CI) = 0.15 (0.08 to 0.3), p value = 0.001), and data collection time (regression coefficient (95% CI) = 0.1 (0.006 to 0.2), p value = 0.04) were the independent predictors for overall QoL scale score [42].

**MLT.** Aruta et al. (2022) by using MLT questionnaire, the results of the path analysis indicated a good data-model fit: ($\chi 2 = 4.97$, df = 2, p = 0.08; CFI = 0.99, TLI = 0.96, SRMR = 0.02, RMSEA [90% CI] = 0.06 [0.000 − 0.13]). The direct effects of safety at home (B = −0.27, $\beta$ = −0.21, SE = 0.05, p ≤ 0.001), TPIs (B = −0.19, $\beta$ = −0.27, SE = 0.05, p ≤ 0.001), and financial difficulties (B = 0.15, $\beta$ = 0.18, SE = 0.05, p ≤ 0.001) on psychological distress were found to be significant. Direct effects of safety at home (B = 0.19, $\beta$ = 0.22, SE = 0.05, p ≤ 0.001), TPIs (B = 0.18, $\beta$ = 0.27, SE = 0.04, p ≤ 0.001), financial difficulties (B = −0.15, $\beta$ = −0.21, SE = 0.05, p ≤ 0.001), and psychological distress (B = −0.29, $\beta$ = −0.34, SE = 0.04, p ≤ 0.001) on quality of life were found to be significant. Results indicated that psychological distress partially mediated the positive influence of safety at home (B = 0.06, $\beta$ = 0.07, SE = 0.02, p ≤ 0.001) and TPIs (B = 0.06, $\beta$ = 0.09, SE = 0.02, p ≤ 0.001) on quality of life [34]. These findings indicate that psychological distress is a mechanism that can partly explain why socio- ecological factors (i.e., safety at home, financial difficulties, and trust in institutions) impact the quality of life of Filipino adults during COVID-19.

## Discussion

Findings of included studies demonstrated how COVID-19 pandemic reduced the QoL of the general population. Different factors influenced directly or indirectly the change of QoL. Researchers utilized different quality of life measurement scales among them EQ-5D leading the pool of measurement scales followed by WHOQoL-BREF then SF12/SF-8/ SF36 as 3[rd] scale and the rest. For studies that used EQ-5D to assess the impact of quality of life, all five dimensions (mobility, self-care, usual activities, pain/discomfort and anxiety/depression) were affected significantly with a mean EQ-5D score of 0.89 with 95% CI [-1.865–2.048] with the lowest score of 0.79 at 95% CI (NR) and upper score of 0.99 at 95% CI (NR) [25]. The mean WHOQoL-BREF score was estimated at 50.55 with a 95% CI [32.19, 68.90]. Other instruments such as SF12 scored 70.60 with 95% CI [57.5, 83.7], SF8 scored QoL at 75.3 with 95% CI [58.7, 91.9] and SF36 (score NR). In low and middle income countries (LMICs) such as Morocco [25] using EQ-5D reported low QoL during confinement as compared to before in the 5 health dimensions respectively; mobility 87%(87%), self-care 97%(93%), usual activities 82%(89%), pain/discomfort 70%(78%) and anxiety/depression 44%(66%) with average QoL at 0.91 (p<0.001) before and 0.86 (0.001) after confinement. Whereas in high income countries (HICs), Belgium for example using EQ-5D before COVID-19 QoL measured 0.82 (95% CI; 0.80–0.84) and during COVID-19 measures 0.79 (95% CI; 0.77–0.81), the same with Netherlands, before COVID-19 EQ-5D measured 0.85 (95% CI; 0.83–0.87) before and during COVID-19 0.84 (95% CI; 0.82–0.86). A research in China, using EQ5D concluded that the mean EQ-5D score and VAS were 0.99 before COVID-19 and 93.5 during COVID-19. When compared HICs and LMICs, both countries were affected significantly by COVID-19 and this was exacerbated by confinement [51]. These results are in line with those of a Chinese study with an average score EQ-5D of 0.949 and VAS score 85.52 [22]. Nine published papers assessed QoL using WHOQoL [7, 30–33, 35, 37, 39, 41] and their mean score was 50.55% with 95% CI [32.19–68.90]. The lower the score, the lower the quality of life. On the other hand, using EQ-5D, the mean score was estimated at 0.89 with 95% CI [0.66–1.13] with the same trend, the lower the score, the lower the quality of life. Our study findings are different from

those published in Vietnam that reported EQ5D score 0.95 (SD = NR.) during the national social distancing, against our results (mean EQ5D = 0.89) [27]. This might be because it is an empirical study while our study summarizes results from a variety of studies making our mean score low. Our main findings rely most on EQ5D and WHOQoL instrument reports. Although, we assessed QoL of the general population during COVID-19 (Mean EQ5D = 0.89), some authors assessed the impact of some chronic diseases on QoL of the general population such as type 2 diabetes [52] (EQ5D = 0.8 SD = 0.20), human immunodeficiency virus (HIV) [53] (EQ5D = 0.8 SD = 0.2), skin disease [54] (EQ5D = 0.73 SD = 0.19), respiratory diseases (EQ5D = 0.66 SD = 0.31), dengue fever (EQ5D = 0.66 SD = 0.24), frail elderly in Vietnam [55] (EQ5D = 0.58 SD = 0.20), elderly after fall injury and facture injuries (EQ5D = 0.46 SD = NR). QoL in general population during COVID-19 was comparable to that of type2 diabetes and HIV. This may be because Type2 diabetes and HIV are chronic conditions, patients are stable on medication if the management and compliance to medications is respected. QoL of skin disease patients, respiratory diseases, dengue fever, frail elderly, elderly after fall and fracture injuries were low as compared to COVID-19 general population. This may be due to the high score of pain involved in these conditions. Different factors that contributed to low quality of life have been identified; age, gender, education level, marital status, financial constraints, confinement, fear of being contaminated and individual with other chronic conditions. The two measurement scales were the most utilized instruments as compared to the other scales and their results show a considerable reduced quality of life. Using WHOQoL-BREF [7] reported a quality of life affected with a score of 39% (CI = NR) and according to authors, males were more affected probably because in developing countries, males are responsible of financial support to the family and because of that, they may fear either confinement that affects job market or else being contaminated and not able to work for their families. Concerning age, participants aged 26-35years were more affected and the reason may be because most peoples of this age bracket are the young couples or single mothers therefore the young fathers were worried about their families and finances if they are quarantined. Females were more affected than their counterparts according to [25] this may be due to the fact that females naturally are the nuclear parts of a family and their emotions towards the family therefore become much worried than males. Widowers had the worst quality of life and this may be due to their worries about their life and that of their children with less psychological support [56, 57] from their spouses. Individuals with chronic diseases (hypertension, Type2 diabetes, asthma, stress, anxiety, depression, etc. . .) had a high risk of low quality of life and this might be because they are vulnerable to COVID-19 with high fear of contamination therefore pushing them to low quality of life. Other factors such as confinement, financial constraints, fear of being contaminated with COVID-19 and having a contaminated family member increased the likelihood of anxiety, stress and depression therefore leading to the low individual quality of life [58]. The main reason of stress due to confinement is due to financial constraint because a confined person is not allowed to work and generate income to sustain the family during the pandemic. It is surprising that both low and high income countries were affected by COVID-19 reducing their population quality of life. This shows how no country in the world was prepared for any huge health pandemic whether rich or poor. This highlights the low level of preparedness for countries to face similar catastrophic situations. What is lacking? Is it the money or strategies? Developed countries can afford to provide necessary means to fight against pandemics but there is no guaranty to protect the populations from dying before actions are in place. For this purpose, there is a need to strengthen infectious disease predictions and modeling using machine learning or artificial intelligence. There is a need to embrace and exploit artificial intelligence to improve the prediction of future events to prevent populations from diseases and death and maintain their maximum quality of life.

## Strengths and weaknesses

First and the foremost, the strength of this review is that, it was conducted according to the international guidelines for systematic reviews after registration of the protocol in international database PROSPERO. Secondly, it was conducted two and half years after the pandemic begun and authors already have published enough papers to allow robust systematic synthesis of results. And the results can be generalized as papers were searched Worldwide with a reasonable sample size (22,967 participants).

There were also some limitations; We searched papers in English only leaving probably out some studies. The fact that we searched only 3 databases and a search engine, some articles might have been missed. The generalizability should be done with caution. Most studies reported the mean QoL during COVID-19 with no baseline to compare, this can weaken our results. All studies were cross-sectional and there were no cohort or case control studies, this can also weaken our conclusions.

## Conclusion

This systematic review confirms that the COVID-19 pandemic affected negatively health related quality of life of the general population. Several factors influencing quality of life of general population through COVID-19 have been identified; age, sex, marital status, education, peoples living with chronic diseases, confinement and financial constraints among others, etc.. . .. There was no significant difference between the impact of COVID-19 in general population in high income countries and low and middle income countries. Three quality of life scales were mainly used to assess the quality of life of the general population; WHO-QoL-BREEF, EQ-5D, SF and others. The findings of this review will be useful for policy makers and health managers to facilitate the planning and prevention of quality of life of the general population during future pandemics. We recommend cohort and case control studies on impact of COVID-19 on quality of life to collect more and strong evidence on impact of COVID-19 on different population in the world. We are also recommending studies on prediction and modeling of infectious diseases using machine learning and artificial intelligence to prevent the population from future pandemics to maintain the population quality of life.

## Supporting information

**S1 Checklist. PRISMA checklist.**
(DOC)

**S1 Table. Summary of included studies.**
(DOCX)

**S1 Text. PubMed search string.**
(DOCX)

## Acknowledgments

The authors acknowledge the moral support from their families, friends and colleagues.

## Author Contributions

**Conceptualization:** Desire Aime Nshimirimana, Donald Kokonya.

**Data curation:** Desire Aime Nshimirimana, Donald Kokonya, Jesse Gitaka.

**Formal analysis:** Desire Aime Nshimirimana, Bernard Wesonga.

**Methodology:** Desire Aime Nshimirimana, Donald Kokonya, Jesse Gitaka, Bernard Wesonga, Japheth Nzioki Mativo, Jean Marie Vianey Rukanikigitero.

**Supervision:** Desire Aime Nshimirimana.

**Writing – original draft:** Desire Aime Nshimirimana.

**Writing – review & editing:** Donald Kokonya, Jesse Gitaka, Bernard Wesonga, Japheth Nzioki Mativo, Jean Marie Vianey Rukanikigitero.

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
