## [Decision Letter · Decision Letter 0]

15 Jun 2023

PGPH-D-23-00668

Impact of COVID-19 on Health-Related Quality of Life in the General Population: A Systematic Review

Dear Dr. Nshimirimana,

Thank you for submitting your manuscript to PLOS Global Public Health. After careful consideration, we feel that it has merit but does not fully meet PLOS Global Public Health’s publication criteria as it currently stands. Therefore, we invite you to submit a revised version of the manuscript that addresses the points raised during the review process.

We look forward to receiving your revised manuscript.

Kind regards,

Anil Gumber, Ph.D.

Academic Editor

Journal Requirements:

1. Please identify your study as "systematic review and meta-analysis" in the title.

2. Please provide separate figure files in .tif or .eps format only and remove any figures embedded in your manuscript file. Please also ensure all files are under our size limit of 10MB.

Additional Editor Comments (if provided):

An Interesting topic is covered by the authors. The paper need thorough English Editing. Abstracts also need better organisation. Aims and objectives of the papers are not clearly highlighted. Methods section need further attention as authors didn't mention inclusion and exclusion criteria. I am really concerned about the measurement of "impact" of COVID on QoL. It seems hardly any paper computed QoL at two points in time (before and during/after COVID) to see changes in QoL scores and then make judgement about the impact of COVID. In my view it just collated information from some cross-sectional studies done before or after COVID. Thus the authors can't justify their title of impact of COVID on QoL by reviewing cross-sectional studies. Therefore authors need to re-write a section separately on those where before and after QoL scores were computed and shown changes in Mean QOL score at two points in time. Table 1 should also mention Change in Mean QoL. Scores in Table 2 are presented but without much explanation about low/high quality of papers. Authors need to mention reasons for including the low quality papers in the review. Discussion is written poorly. It just summarises findings and needs to focus also on changes in QoL as a result of COVID.

Reviewers' comments:

Reviewer's Responses to Questions

**Comments to the Author**

1. Does this manuscript meet PLOS Global Public Health’s publication criteria? Is the manuscript technically sound, and do the data support the conclusions? The manuscript must describe methodologically and ethically rigorous research with conclusions that are appropriately drawn based on the data presented.

Reviewer #1: Yes

Reviewer #2: Yes

2. Has the statistical analysis been performed appropriately and rigorously?

Reviewer #1: Yes

Reviewer #2: Yes

3. Have the authors made all data underlying the findings in their manuscript fully available (please refer to the Data Availability Statement at the start of the manuscript PDF file)?

Reviewer #1: Yes

Reviewer #2: Yes

4. Is the manuscript presented in an intelligible fashion and written in standard English?

Reviewer #1: Yes

Reviewer #2: Yes

5. Review Comments to the Author

Reviewer #1: I have the following comments for the authors to address. I am happy to review this paper again.

1) The important finding for this study is " EuroQuality of Life-Five dimensions was 0.88

95% confidence interval [0.69-1.07]. ". It is important to guide authors how does 0.88 compare to other disorders. Please compare with the following conditions and discuss in the discussion so that readers know where it stands:

General population under COVID-19 (EuroQol-5D = 0.95) (Tran et al 2020),

patients suffering from diabetes (EuroQol-5D= 0.8) Nguyen Huong Thi Thu et al 2018),

human immunodeficiency virus (HIV) (EuroQol-5D = 0.8) (Tran et al 2018),

skin diseases (EuroQol-5D= 0.73) (Nguyen et al 2019),

respiratory diseases (EuroQol-5D= 0.66) (Ngo et al 2019),

dengue fever (EuroQol-5D= 0.66) (Tran et al 2018),

frail elderly (EuroQol-5D = 0.58) (Nguyen Anh Trung et al 2019)

elderly after fall injury (EuroQol-5D = 0.46) (Vu et al 2019)

fracture injuries (EuroQol-5D = 0.23) (Vu et al 2019).

References:

Tran BX, Nguyen HT, Le HT et al. Impact of COVID-19 on Economic Well-Being and Quality of Life of the Vietnamese During the National Social Distancing. Front Psychol. 2020 Sep 11;11:565153. doi: 10.3389/fpsyg.2020.565153. PMID: 33041928; PMCID: PMC7518066.

Nguyen, Huong Thi Thu et al. “Health-related quality of life in elderly diabetic outpatients in Vietnam.” Patient preference and adherence vol. 12 1347-1354. 27 Jul. 2018, doi:10.2147/PPA.S162892

Tran, Bach Xuan et al. “Depression and Quality of Life among Patients Living with HIV/AIDS in the Era of Universal Treatment Access in Vietnam.” International journal of environmental research and public health vol. 15,12 2888. 17 Dec. 2018, doi:10.3390/ijerph15122888

Nguyen, Sau Huu et al. “Health-Related Quality of Life Impairment among Patients with Different Skin Diseases in Vietnam: A Cross-Sectional Study.” International journal of environmental research and public health vol. 16,3 305. 23 Jan. 2019, doi:10.3390/ijerph16030305

Ngo, Chau Quy et al. “Effects of Different Comorbidities on Health-Related Quality of Life among Respiratory Patients in Vietnam.” Journal of clinical medicine vol. 8,2 214. 7 Feb. 2019, doi:10.3390/jcm8020214

Tran, Bach Xuan et al. “Cost-of-Illness and the Health-Related Quality of Life of Patients in the Dengue Fever Outbreak in Hanoi in 2017.” International journal of environmental research and public health vol. 15,6 1174. 5 Jun. 2018, doi:10.3390/ijerph15061174

Nguyen, Anh Trung et al. “Frailty Prevalence and Association with Health-Related Quality of Life Impairment among Rural Community-Dwelling Older Adults in Vietnam.” International journal of environmental research and public health vol. 16,20 3869. 12 Oct. 2019, doi:10.3390/ijerph16203869

Vu, Hai Minh et al. “Effects of Chronic Comorbidities on the Health-Related Quality of Life among Older Patients after Falls in Vietnamese Hospitals.” International journal of environmental research and public health vol. 16,19 3623. 27 Sep. 2019, doi:10.3390/ijerph16193623

Vu HM, Dang AK, Tran TT, et al Health-Related Quality of Life Profiles among Patients with Different Road Traffic Injuries in an Urban Setting of Vietnam. Int J Environ Res Public Health. 2019 Apr 24;16(8):1462. doi: 10.3390/ijerph16081462. PMID: 31022979; PMCID: PMC6517995

2) The authors should discuss the impact of COVID-19 on burnout, depression and anxiety in the introduction, that may affect quality of life. Please discuss the following findings:

Search PubMed for: Burnout is an important public health issue at times of the COVID-19 pandemic. Current measures which focus on work-based burnout have limitations in length and/or relevance. When stepping into the post-pandemic as a new Norm Era, the burnout scale for the general population is urgently needed to fill the gap. This study aimed to develop a COVID-19 Burnout Views Scale (COVID-19 BVS) to measure burnout views of the general public in a Chinese context and examine its psychometric properties.

A systematic review of COVID-19 on mental health

Search for PubMed: Relatively high rates of symptoms of anxiety (6.33% to 50.9%), depression (14.6% to 48.3%), post-traumatic stress disorder (7% to 53.8%), psychological distress (34.43% to 38%), and stress (8.1% to 81.9%) are reported in the general population during the COVID-19 pandemic in China, Spain, Italy, Iran, the US, Turkey, Nepal, and Denmark.

The impact of COVID-19 on three continents and its relationship with physical health:

Search for PubMed: The results showed that Poland and the Philippines were the two countries with the highest levels of anxiety, depression and stress; conversely, Vietnam had the lowest mean scores in these areas. Chain mediation model showed the need for health information, and the perceived impact of the pandemic were sequential mediators between physical symptoms resembling COVID-19 infection (predictor) and consequent mental health status (outcome).

3) Under the method, the authors stated "A random-effects model was fitted to the data." Please provide more information about random-effects model:

Tandom-effects model attempted to generalize findings beyond the included studies by assuming that the selected studies are random samples from a larger population (Cheung et al 2012).

Reference:

Cheung MW et al (2012) Conducting a meta-analysis: basics and good practices. Int J Rheum Dis. 2012 Apr;15(2):129-35. doi: 10.1111/j.1756-185X.2012.01712.x. Epub 2012 Feb 22. PMID: 22462415.

Reviewer #2: This contribution provides the results of a systematic review of papers published in English language on associations of quality of life with COVID -19, based on 25 studies retrieved from different countries. Methodologically, the search procedure and data analysis were done according to international quality standards, with a pre-published protocol and following PROSPERO criteria, assessing risk of bias and heterogeneity as well as study quality check. The different measurement tools are adequately described, and main study findings are enumerated in text and tables. There is no surprise that, overall, reduced quality of life is observed in association with COVID-19. In its current version, the manuscript suffers from several shortcomings listed here.

1. In Figure 1, 1.700.074 hits are reported from first search strategy. Although search terms include ‘quality of life’, authors report that this large primary number of titles was reduced to 546. Information on how this most substantial reduction was achieved is missing.

2. Background: At the end of this Introduction authors should state what exactly the aims of their review are. For instance, they could mention that they are interested in examining the consistency of findings across different measurement approaches, or to test risk of bias in this field of research, or to explore variations of quality of life according to sociodemographic variables etc.

3. Methods: On p. 9, information on measurements in different countries is given twice on the same page.

4. Results: In Table 1, in the arrow ‘mean quality of life’ percentage is given in several cases. Please explain. Moreover, information about means in different studíes cannot be interpreted without additional information on the scale range and some standard value in healthy, unaffected populations. In contrast to Table 1, information given in Table 2 is very useful and well done.

5. Discussion: The section on study limitations needs critical extension. The relatively poor study quality (mainly cross-sectional!) needs to be emphasised. Authors should also make clear how data on comparisons of quality of life before and during pandemic were recorded in the frame of cross-sectional studies (Ref: 19, 22, 40, 41).

6. PLOS authors have the option to publish the peer review history of their article (what does this mean?). If published, this will include your full peer review and any attached files.

**Do you want your identity to be public for this peer review?** For information about this choice, including consent withdrawal, please see our Privacy Policy.

Reviewer #1: No

Reviewer #2: **Yes: **Johannes Siegrist

<quillbot-extension-portal></quillbot-extension-portal><quillbot-extension-portal></quillbot-extension-portal>

---

## [Editor Report · Decision Letter 1]

4 Oct 2023

Impact of COVID-19 on Health-Related Quality of Life in the General Population: A Systematic Review and Meta-analysis

PGPH-D-23-00668R1

Dear Dr Nshimirimana,

We are pleased to inform you that your manuscript 'Impact of COVID-19 on Health-Related Quality of Life in the General Population: A Systematic Review and Meta-analysis' has been provisionally accepted for publication in PLOS Global Public Health.

Best regards,

Anil Gumber, Ph.D.

Academic Editor

thanks for incorporating reviewers suggestions.